



# Satellite rainfall products outperform ground observations for landslide forecasting in India

Maria Teresa Brunetti[1], Massimo Melillo[1], Stefano Luigi Gariano[1], Luca Ciabatta[1], Luca Brocca[1], Giriraj Amarnath[2], Silvia Peruccacci[1]

[1]IRPI, via Madonna Alta 126, 06128, Perugia, Italia
[2]International Water Management Institute, Colombo, Sri Lanka

*Correspondence to*: Maria Teresa Brunetti (maria.teresa.brunetti@irpi.cnr.it)

**Abstract.** Landslides are among the most dangerous natural hazards, particularly in developing countries where ground observations for operative early warning systems are lacking. In these areas, remote sensing can represent an important tool to

forecast landslide occurrence in space and time, particularly satellite rainfall products that have improved in terms of accuracy and resolution in recent times. Surprisingly, only a few studies have investigated the capability and effectiveness of these products in landslide forecasting, to reduce the impact of this hazard on the population.

We have performed a comparative study of ground- and satellite-based rainfall products for landslide forecasting in India by using empirical rainfall thresholds derived from the analysis of historical landslide events. Specifically, we have tested Global

Precipitation Measurement (GPM) and SM2RAIN-ASCAT satellite rainfall products, and their merging, at daily and hourly temporal resolution, and Indian Meteorological Department (IMD) daily rain gauge observations. A catalogue of 197 rainfall-induced landslides occurred throughout India in the 13-year period between April 2007 and October 2019 has been used.

Results indicate that satellite rainfall products outperform ground observations thanks to their better spatial (10 km vs 25 km) and temporal (hourly vs daily) resolution. The better performance is obtained through the merged GPM and SM2RAIN-

ASCAT products, even though improvements in reproducing the daily rainfall (e.g., overestimation of the number of rainy days) are likely needed. These findings open a new avenue for using such satellite products in landslide early warning systems, particularly in poorly gauged areas.

## 1      Introduction

In India, the annual toll in terms of human lives and loss of property, due to landslides and floods, urgently demands the

implementation of long-term strategies to prevent hydrogeological instability. India is the country with the highest number of non-seismically triggered disastrous landslides as per the global dataset published by Froude and Petley (2018), with more than 600 records and 16% of catalogued rainfall-induced landslides. According to the EM-DAT disaster database (Guha-Sapir et al., 2009), a total of 48 landslide events have occurred since 1950 with 4,535 deaths and 385,000 people affected. In recent years, the usability of regional and local landslide early warning systems using forecasting of rainfall-induced landslides is an

approach now adopted in various countries to mitigate landslide risks (Guzzetti et al., 2020). When heavy and/or abundant



rainfall combines with a high landslide susceptibility, an operational warning system is the best solution to mitigate the impact of the event. The use of rainfall thresholds, combined with susceptibility maps, makes it possible to estimate the probability of landslide triggering (Rossi et al., 2018). Rainfall thresholds all over the world are derived from the statistical analysis of rainfall conditions that have resulted in past landslides (e.g., Guzzetti et al., 2007, 2008; Cepeda et al. 2010; Sengupta et al., 2010;

Ruiz-Villanueva et al. 2011; Berti et al., 2012; Staley et al., 2013; Zêzere et al., 2015; Rosi et al., 2016; Peruccacci et al., 2017; Segoni et al., 2018; Gariano et al., 2019; Valenzuela et al.,  2019; Jordanova et al., 2020; Leonarduzzi and Molnar, 2020; Salinas-Jasso et al., 2020; Uwihirwe et al., 2020). Defining rainfall thresholds over large and diverse areas needs statistically significant information on the occurrence of past landslides and a large amount of rainfall data. On the other hand, India with diverse geographical region with varied climate, complex topography and geomorphology does not allow such a density of

rainfall stations as to accurately reconstruct the rainfall that caused the landslides. To make a comparison, in Italy the average density of rain gauges is about $1/130$ km$^{-2}$ (Peruccacci et al., 2017), while in India it is about $1/470$ km$^{-2}$ (6995 stations over $3.287.000$ km$^2$) (Pai et al., 2014). In addition, in remote areas (e.g., mountainous terrain) of India the low density of rain gauges penalizes the representativeness of the measurements. Hence, the use of satellite rainfall products is a valid option for measuring precipitation over such large areas, provided that a performance analysis and assessment in capturing reliable

rainfall measurements are carried out, even in areas characterized by complex orography and severe weather systems (e.g., Gupta et al., 2020; Thakur et al., 2019; Tang et al., 2020).

So far, only a few works have studied the relationship between rainfall and landslides in India, and mainly for limited areas or individual states. The analysis of the rainfall conditions responsible for slope failures were conducted using both data from rain gauges and satellite data. Authors using ground-based measurements include Jaiswal and van Westen (2009) who

calculated rainfall thresholds based on the daily rainfall vs. the 5-days antecedent rainfall along two sections of a historic railway and of a national highway in the state of Tamil Nadu, southern India. Kanungo and Sharma (2014) derived intensity-duration (*ID*) thresholds based on daily rain gauge measured rainfall along a stretch of a national highway of Garhwal Himalayas in the Uttarakhand State, northern India. More recently, Naidu et al. (2018) identified rainfall threshold based on the daily rainfall vs. the 2-, 3- and 5-days antecedent rainfall and in a small hamlet in Kerala, Southwest India. One-dimensional

probabilistic thresholds were defined by Dikshit and Satyam (2019) for a small ($\sim5$ km$^2$) area in Darjeeling Himalayas, north eastern India. In the same area, Dikshit et al. (2020) identified their best threshold based on the daily rainfall vs. the 20-days antecedent rainfall. *ID* regional and local rainfall were derived by Geethu et al. (2019) in north eastern India for the Sikkim region and the Gangtok area, respectively. More recently, Mandal and Sarkar (2020) estimated rainfall thresholds along a vulnerable section of the NH10 road in Darjeeling Himalayas.

Few authors used satellite-based data to explore the correlation between rain and the occurrence of landslides. Mathew et al. (2014) obtained and validated *ID* thresholds using the Tropical Rainfall Measurement Mission (TRMM) precipitation estimates in a small area of Garhwal Himalayas in the Uttarakhand State, northern India. They also applied a logistic regression model to assess the effect of the antecedent rainfall using rain gauge data. Thakur et al. (2020) analysed the rainfall triggering three





large landslides over Western Ghats, southern west India. They used TMPA (product of TRMM) and IMERG (Integrated
Multi-satellitE Retrievals for GPM, Global Precipitation Measurement) derived rainfall data. In particular, the performance of
IMERG V5 was assessed during the awful Malin landslide, which occurred in 2014, at the initial stage of the GPM product.

The above review shows that the prediction of rainfall-induced landslides in India relies mostly upon custom approaches
applied to local areas, being a comprehensive method still missing at a larger regional scale. Conversely, at the global scale
Hong et al. (2006) evaluated the potential of the real-time TMPA product to assess its predictive ability for rainfall-triggered
landslides. Recently, Kirschbaum and Stanley (2018) proposed a model to provide the potential landslide activity combining
satellite-based precipitation estimates (IMERG from GPM) with a landslide susceptibility map. Moreover, examples of
application of various satellite-based rainfall estimations for the definition of rainfall thresholds for landslide forecasting at a
national scale were proposed by Robbins (2016) in Papua New Guinea, Brunetti et al. (2018a) in Italy, He et al. (2020) in
China, and by Monsieurs et al. (2019) over the East African Rift.

However, a more in-depth and dedicated focus on the use of the available rainfall products at the regional scale is required to
best predict the spatial and temporal occurrence of landslides in India. For this purpose, we collected in a catalogue the
information on landslides initiated by rainfall in India. We investigated the role of the rainfall in the occurrence of landslides
using empirical rainfall thresholds (Peruccacci et al., 2017 and references therein), and we compared the performance of
satellite-based and ground-based rainfall products. Comparing the thresholds with rainfall measures, estimates, and forecasts
is the basis of most current operational landslide early warning systems (Guzzetti et al., 2020).

## 2      Study area

India is the seventh largest country in the world and forms a well-defined peninsula of Asia, bounded to the North by the
imposing mountain chain of the Himalayas and surrounded by the Arabian Sea to the West and the Bay of Bengal to the East,
respectively (Figure 1).

Three basic structural units are usually recognized in the land: the Himalayas in the North, the peninsular Deccan plateau in
the south and the Indo-Gangetic Plain between the two. The country also includes two groups of islands, Lakshadweep, in the
Arabian Sea, and the Andaman and Nicobar Islands, between the Bay of Bengal and the Andaman Sea. Apart from the highest
mountains in the Himalayas, the main reliefs of the Deccan plateau region are the Western Ghats, a North-South chain of
mountains or hills in the western edge, and the Eastern Ghats, running mostly in a Northeast-Southwest direction. The steep
topography of the diverse regions and their geological complexity produce an intense landslide activity, especially during the
monsoon season.

The climate of Southeast Asia is dominated by the monsoon for over one-third of the year (Annamalai et al., 1999). Neal et al.
(2019) identified seven broad-scale weather regimes in India. Among them, the main monsoon season occur from June to
September, being mostly active during July and August. The monsoon is associated to heavy and persistent rain starting
generally on the 1st or 2nd June from the south-west coast of India (Mooley and Shukla, 1987; Rao et al., 2005) and extending





across central and northern areas, reaching Mumbai around 10th June (Adamson and Nash, 2013) and north-west India by 15th July on average (Tyagi et al., 2011). The large amount of rain in this period makes this the part of the year when the most landslides occur in India. Additionally, non-monsoonal precipitation patterns, the western disturbances, may occur even in January and February, bringing rain or snow to north western parts of India later moving to north-eastern parts in early spring.

**3      Data and methods**

In order to study the relationship between landslides and rainfall, it is necessary to find information on a statistically significant number of slope failures (Peruccacci et al., 2012). It is necessary, as well, sufficiently long and continuous series of rainfall data from ground- and/or satellite-based observations. Landslide have to be geo-localized with adequate accuracy, and at least the occurrence day must be known (Brunetti et al., 2018b). Similarly, the rainfall series should have a sufficient temporal
resolution to make the measurements representative of the rainfall conditions that have presumably triggered the slope failures. Generally, the availability of data for such events depends on the availability and accessibility of local information sources. In India, given the vastness and ecological diversity of the territory, the collection of accurate landslide information is often a challenge. The lack of detailed information associated with the rapid evolution of the landscape due to the high intensity and extreme rainfall often makes it difficult if not impossible locating the slope failures. In case of incomplete or missing data, a
landslide event cannot be included in the analysis and it is therefore not included in the catalogue. Following these criteria, we discarded nearly 50% of the collected rainfall-induced landslide events.

**3.1 Landslide data**

We compiled a catalogue of 197 rainfall-induced landslides occurred in India in the 13-year period between April 2007 and October 2019. The geographical distribution of the slope failures is shown in Figure 1, which reveals how landslides are
basically clustered in four main areas corresponding to the main mountain reliefs. One area is in the north eastern part of India and includes West Bengal (30 landslides), Sikkim (12), Assam (21), Manipur (10), Arunachal Pradesh (5), Meghalaya (3), Nagaland (2) and Mizoram (2); another area in the north western part encompasses Uttarakhand (9), Himachal Pradesh (8), Jammu and Kashmir (7) and Rajasthan (1); the third area in the western coast encloses Kerala (50), Karnataka (18), Maharashtra (6), Tamil Nadu (6) and Goa (1); finally, 2 landslides are located near the eastern coast in Andhra Pradesh.
Landslides collected in the catalogue are mainly located in inhabited areas and along communication routes. Likely for this reason, about 25% of them have caused one or more fatalities (see pie chart in Figure 1) confirming how the social incidence of this type of hazard is relevant in India. A more in-depth analysis shows that the monthly distribution of the slope failures (Figure 2a) is peaked in June, whilst the highest number of fatalities are in August (Figure 2b). The 2010 Ladakh (Jammu and Kashmir) debris flows and mudflows and the 2014 Malin (Maharashtra) landslides substantially contribute to the two peaks in
Fig. 2b.

We gathered information from multiple sources (Figure 3a), such as online newspapers and magazines (ON), blogs (BG), technical reports (TR) made available by the Geological Survey of India (GSI, www.gsi.gov.in), scientific journals (SJ) and social media (SM), e.g. Twitter, Facebook, etc. Most of the data (53.4%) in the catalogue is found in TR. When the information is proven accurate in terms of spatial and temporal location, the landslide is included in the catalogue and is represented as a point on the map (white and red dots in Figure 1). Each landslide in the catalogue is assigned a position accuracy level in four classes (Figure 3b), and a decreasing temporal accuracy in three classes (Figure 3c).

In about half of the cases the lack of data on the landslide type, or the non-technical language used by the source of information make it difficult to determine the type of movement and the involved material, and the landslide is generically classified as "not specified" (NS). In other cases, the identification of the landslide type is allowed by photographs or by the assessments of technical personnel (as for surveys carried out by the GSI). As a result, the collected information on the landslide type has been grouped in five classes: complex landslide (CL), debris-flow (DF), earth or mud flow/slide (EF), rock fall/slide (RF), and deep-seated landslide (DL). Here it is stated that this classification follows a criterion that gives greater priority to the type of material (earth/mud or rock) rather than the type of movement (flow, slide or fall). Figure 4 shows the spatial distribution of landslides classified by type, and reveals that about half of the landslides are not specified (NS, 49.7%), whereas the remaining are mostly EF (24.4%), RF (17.1%) and DF (5.7%). CL and DL account each for a smaller percentage ($\leq$ 2.1%). In particular, most of the RF are located in the Himalayas, and the remaining are scattered throughout the rest of the territory.

## 3.2 Rainfall data

The rainfall measurements are obtained through three datasets available over the study area. In order to provide a robust and thorough analysis, we used observations obtained through ground stations as well as satellite sensors. The following assumptions apply to each of the considered datasets:

1) the rainfall estimate is used at its own native spatial resolution;
2) the analysis has been carried out at hourly and daily temporal scale; if the resolution was higher, the data have been summed up in order to obtain hourly rainfall.

The Indian Meteorological Department (IMD) dataset is based on ground observations coming from more than 6900 rain gauges (Pai et al., 2014). The rainfall recorded at each station is interpolated in space by using an inverse distance weighted algorithm, providing a daily gridded dataset with a spatial resolution of 0.25° (25 km) over the territory of India. Hereinafter this dataset is referred as IMD.

SM2RAIN-derived rainfall is obtained by applying the SM2RAIN algorithm (Brocca et al., 2014) to satellite-based soil moisture data. By inverting the soil water balance equation, the algorithm allows to estimate rainfall directly from soil moisture observations. The algorithm has proven to provide accurate and reliable rainfall estimates that have been already used for natural hazard evaluation studies (Ciabatta et al., 2017, Brunetti et al., 2018a, Camici et al., 2020) ensuring satisfactory results. In this study, rainfall data obtained through the application of SM2RAIN to ASCAT satellite soil moisture product (Wagner et al., 2013) provided by H SAF (the EUMETSAT Satellite Application Facility on Support to Operational Hydrology and





Water Management) as product H117 is used. The product, hereinafter referred as SM2R, has a spatial resolution of 0.1° (12.5
km) and a daily temporal resolution.

The Integrated Multi-satellitE Retrievals for GPM (IMERG, Huffman et al., 2018) Early Run is used in this study as state-of-
the-art satellite rainfall product. The dataset is obtained by running the algorithm at 0.1° (10 km) spatial and at half-hourly
temporal resolution from a constellation of microwave and infrared satellites. The Early Run version is characterized by a
latency of 4-6 hours after sensing. In order to achieve the hourly temporal resolution used for the analysis, two subsequent
rainfall data are summed up. Hereinafter this dataset is referred as IMERG-ER.

In order to test and highlight the added value of integrating rainfall estimates obtained through different approaches for
landslides forecasting, two additional merged products obtained through the merging of SM2R and IMERG-ER are created
and used as input for determining the rainfall thresholds. The integration between these two different products has already
been tested satisfactorily (Ciabatta et al., 2017; Massari et al., 2020). The integration allows to take benefit of the capabilities
of each approach and to limit the drawbacks, i.e. underestimation of rainfall by SM2RAIN when the soil is close to saturation
or the overestimation by IMERG-ER for low-intensity rainfall events. The integration has been performed by following the
approach proposed in Ciabatta et al. (2017). For the first merged product IMERG-ER is summed up in order to obtain a daily
temporal resolution. The merging between the two products is obtained by using Eq. (1):

$$S_{merged} = S_{SM2R} + w_i(S_{IMERG-ER} - S_{SM2R}) \,, \tag{1}$$

where $w_i$ is the integration weight, ranging from 0 to 1, and it is estimated for each pixel using Eq. (2) (Kim et al., 2015):

$$w_i = \frac{\rho_{SM2R-R} - (\rho_{IMERG-ER,SM2R} \cdot \rho_{IMERG-ER,R})}{\rho_{IMERG-ER,R} - (\rho_{IMERG-ER,SM2R} \cdot \rho_{SM2R,R}) + \rho_{SM2R,R} - (\rho_{IMERG-ER,SM2R} \cdot \rho_{IMERG-ER,R})} \,, \tag{2}$$

where $\rho_{S1,S2}$ is the correlation between two generic datasets S1 and S2 and R is a reference rainfall product. In this analysis,
ERA5 (Hersbach et al., 2020) is chosen as reference rainfall product for estimating the integration weights. The integration
between IMERG-ER and SM2R is the PMERG-D (daily) dataset.

The second merged product is obtained taking the total amount of rainfall estimated by PMERG-D and distributing it within
each day by considering the hourly temporal distribution of IMERG-ER. In this way, the SM2R rainfall at the hourly temporal
resolution is obtained. Hereinafter this dataset is referred as PMERG-H.

Both PMERG-D and PMERG-H have the native spatial resolution of SM2R, i.e., 0.1° (12.5 km).

### 3.3 Reconstruction of the triggering rainfall

The quantitative reconstruction of the rainfall conditions that likely caused the observed landslides is the first step for the
identification of effective rainfall thresholds. We used CTRL-T software tool proposed by Melillo et al. (2018) to single out
rainfall events starting from continuous rainfall series. For each landslide, the tool: 1) selects automatically the representative
pixel; 2) identifies the most probable duration and cumulated rainfall (D,E) conditions assumed to have caused the landslides
collected in the catalogue; 3) calculates empirical cumulated event rainfall-rainfall duration (ED) thresholds at various non-





exceedance probability (NEP) adopting the frequentist approach and a bootstrap technique (Brunetti et al., 2010; Peruccacci et al., 2012).

CTRL-T works on a continued series of hourly or daily rainfall measurements or estimates for each pixel. We found that rainfall records both for the ground-based and the satellite-based products contain some spurious values that prevent the reconstruction of the individual rainfall events by the CTRL-T tool (Melillo et al., 2015). To recognize and remove spurious

events, we calibrated the cumulated rainfall in each pixel at the proper temporal resolution (hourly or daily) using the rainfall series of the Darjeeling District in the driest January and February months as a reference. Analysing hyetographs from ground-based and satellite-based products we applied a filter that removes 2 mm per hour for IMERG-ER and PMERG-H and 2 mm per day for IMD, SM2R and PMERG-D. We are aware that the filter may remove a small percentage (5 mm as a median) of the daily cumulated rainfall, nonetheless this amount is nearly irrelevant for the landslide triggering.

Each landslide is associated to a single pixel of each product. Figure 5a shows a nested view of the cell grids for the ground-based (IMD) and two satellite-based (IMERG-ER and SM2R) products. To increase the reliability of the rainfall estimates all the pixels whose centre falls into a circular buffer of 20 km-radius from the landslide are enclosed in the analysis (Figure 5b). The reason for this choice is that often the pixel where the landslide falls may not faithfully measure the triggering rainfall. Possible causes for this are: (i) malfunctioning of the sensor, (ii) the spatial averaging in the grid cell (e.g., an intense

precipitation limited to a small area around the landslide), (iii) the case of landslides with medium (P2) to low (P4) position accuracy, and/or (iv) low temporal accuracy (T2 and T3). For one or more of the above reasons, CTRL-T is generally not able to reconstruct the triggering rainfall for all the landslides in the catalogue.

To identify the representative pixel to be analysed, we used a weight, $w = E^2 \cdot D^{-1}$ (modified after Melillo et al., 2018). For the pool of pixels enclosed in the 20-km buffer centred on the landslide the representative pixel is the one for which the ($D,E$) pair

provides the highest weight. For a given landslide in the catalogue, Figure 6 portrays the colour associated with the pixels selected for the reconstruction of the triggering rainfall for each product. The darker the colour, the highest the weight and hence the expected representativeness of the pixel. It can be observed that in many cases, the representative pixel (the darkest one) is not necessarily the one containing the landslide (yellow dot).

Once rainfall conditions have been reconstructed, CTRL-T calculates the rainfall thresholds. Empirical *ED* thresholds are

power law curves with the form:

$$E = (\alpha \pm \Delta\alpha) \cdot D^{(\gamma \pm \Delta\gamma)},$$ (3)

where *E* is in mm, *D* is in hours (or in days), $\alpha$ and $\gamma$ are the scale and the slope parameters of the curve, respectively, and $\Delta\alpha$ and $\Delta\gamma$ are the associated uncertainties (Peruccacci et al., 2012). It is worth to notice that the two temporal scales for daily and hourly datasets has to be expressed in days and hours, respectively (Gariano et al., 2020).

In order to assess the performance of the ground- and satellite-based rainfall products in forecasting landslides in India a validation of the thresholds has been carried out, following Brunetti et al., (2018a). For each product, we used 70% of the reconstructed (*D,E*) pairs to generate 100 synthetic rainfall series starting from a random seed. Then, rainfall thresholds at



varying NEP are calculated. The remaining 30% is used to evaluate the threshold performance by means of contingency table (reporting binary classifiers of rainfall conditions that triggered or did not trigger landslides), skill scores and ROC (Receiver

Operating characteristic) analysis, following Gariano et al. (2015) and Brunetti et al. (2018a). In particular POD (Probability of detection) and POFD (Probability of False Detection) are calculated as follows: POD = TP/(TP+FN) and POFD = FP/(FP+TN), where TP is a true positive and FN is a false negative, i.e. a landslide-triggering rainfall condition located above and below the threshold, respectively, while FP is a false positive and TN is a true negative, i.e. a rainfall condition without landslides located above and below the threshold, respectively. POD and POFD are used as y- and x- values of the ROC space,

respectively (Fawcett, 2006). Each (POFD, POD) pair represents the prediction performance of a threshold and the Euclidean distance of the pair from the upper left corner in the ROC plane (POD=1, POFD=0) is used as a measure of the goodness of the threshold. Moreover, ROC curve for each rainfall product is drawn by varying the NEP of the threshold and the area under curve (AUC) is used as a measure of the goodness of each product.

## 4    Results

In the following, we report the analyses of the rainfall measurements and rainfall estimates of the five products to compare their capability to capture the rainfall over India.

Figure 7a shows the box-and-whiskers plots of the monthly rainfall for each product in the analysed time period (2007-2019) for the selected pixels with landslides. With regard to the daily temporal resolution, Figure 7a reveals that the rainfall estimated by PMERG-D is comparable to that of SM2R, as expected for a derived product. The two exhibit median values similar or

higher than the ground-based IMD product especially during the rainy season. At the hourly temporal resolution, the two IMERG-ER and PMERG-H products look noticeably different, being the rainfall from IMERG-ER steadily higher over time. As an example, in the two peak monsoon months of July and August the median value is nearly constant and amounts to about 480 mm for IMERG-ER, while it decreases from 340 mm to 280 mm for PMERG-H. Overall for the two months, the median of IMERG-ER is from 1.4 to 1.6 times higher than that of PMERG-H, while in the dry period this proportion is even more

than six (e.g., November and December). The PMERG-H merged product is comparable to IMD, whereas on average the median value of IMERG-ER is about 1.5 and 2.5 times higher than IMD in the rainy and dry season, respectively. In order to investigate how the rainfall is distributed over a daily scale, we calculated the number of rainy days per month for each product (Fig. 7b).

A variable number of rainy days is observed among the various datasets except for SM2R and PMERG-D, which are very

similar. These two products have constantly the highest number of rainy days, and they show a kind of saturation in July and August when the rainfall turns out to be uninterrupted for the whole month (30 days). Among the satellite-based datasets, IMERG-ER is the one with the lowest number of rainy days, nearly comparable to that of IMD, even though IMERG-ER has the most abundant rainfall (Fig. 7a). Overall, Figure 7b highlights that SM2R and PMERG-D likely overestimate the number





of rainy days, especially from May to October. In particular, in October we obtain median values of 7 rainy days for IMD, 9 for IMERG-ER, 12 for PMERG-H, and 23 for SM2R and PMERG-D.

Once we reconstructed for each product the rainfall $(D,E)$ conditions that have triggered the landslides in the catalogue, we calculated $ED$ rainfall thresholds at varying NEP for the daily and the hourly datasets. As an example, Figure 8a shows the comparison between rainfall $(D,E)$ pairs and relative thresholds at 5% NEP for SM2R and PMERGE-D. From the figure it is evident that SM2R and PMERGE-D $(D,E)$ pairs are shifted towards longer durations than those associated with IMD. The

$T_{5,SM2R}$ threshold is the lowest one and is parallel to $T_{5,IMD}$ . For rainfall duration shorter than 12 days, $T_{5,PMERGE-D}$ is the highest one. Figure 8b compares the thresholds of the two hourly datasets and shows that rainfall $(D,E)$ conditions from IMERG-ER are on average more severe than those reconstructed using PMERG-H data. As a consequence, $T_{5,IMERG-ER}$ turns to be higher than $T_{5,PMERG-H}$.

Table 1 lists for each product the number of landslides, the descriptive statistics of the rainfall $(D,E)$ conditions used to define

the thresholds and the equations of the curves, with their uncertainties. For the daily-based satellite data, the median duration $(D)$ of the triggering rainfall is more than seven times longer than that of IMD, and the maximum duration is about three times, confirming what observed in Figure 8a. Consequently, the median value of the cumulated rainfall for SM2R and PMERG-D is 4.5 to 5.2 times higher than that of IMD. For the hourly-based data, the duration median value for IMERG-ER is higher than that of PMERG-H, whereas the maximum value is about one-half. According to Figure 7a, the median and the maximum

cumulated rainfall $(E)$ for IMERG-ER is largely higher than that of PMERG-H. For an easier comparison between the thresholds, the uncertainties have not been drawn in Figures 8a,b, but are shown in Table 1.

The performance of the ground- and satellite-based rainfall products in forecasting landslides in India is evaluated through the ROC analysis. Figure 9 portrays the comparison between the rainfall products varying the NEP of the threshold, where error bars represent their uncertainties. For a quantitative analysis, the inset graph in the figure shows AUC for each product. Based

on this outcome, the two hourly PMERGE-H and IMERG-ER are performing best in predicting landslides in India, whereas the ground-based IMD is the less performing product. We acknowledge that the comparison may be biased by the different temporal and spatial resolution of the data.

## 5    Discussion and conclusions

The comparative study among different rainfall products in forecasting rainfall-induced landslides have shown the

outperformance of satellite products over ground-based observations in India (Fig. 9). Nevertheless, a detailed analysis of the procedure used and the results obtained is mandatory to infer the actual potential of the satellite rainfall estimations to predict the rainfall conditions that may initiate future landslides. First, we acknowledge that the landslides gathered in the catalogue are a limited sample of the analysed time period for which we found detailed spatial and temporal information (Figs. 3b,c). Nevertheless, we maintain that the dataset is a good sample being most of the information found in technical reports (Fig. 3a)

that usually meet reliability requirements.





Analysing the impact of landslides on the population, Figure 2b shows that the highest numbers of deaths is reported in July and August, while instead the highest number of slope failures occurred in June (Fig. 2a). The lack of a proportion could be ascribed to the abundant and intense rainfall of July and August throughout the territory (Fig. 7a) that is able to initiate extensive, fast landslides (e.g. mudflows) so severe as to destroy homes and villages, causing human losses. We observed that

the landslides in the catalogue for which we know the type and that have caused more victims (82%) are earth or mud flow/slides (EF).

In order to reconstruct the individual rainfall events with the CTRL-T tool, we removed a sort of white noise from data by using the rainfall series of the Darjeeling District in January and February. We acknowledge that such calibration could be dependent on the location where it was performed and may also be dependent on the amount of rain (e.g., in monsoon months).

The rainfall estimated by SM2R and PMERGE-D is almost continuous in the monsoon months as shown by the number of rainy days, considerably higher than that of IMD and of the hourly IMERG-ER and PMERG-H products (Fig. 7b). This is in spite of the median monthly rainfall which is in the same range of that measured or estimated by the other products. Overall, the rainfall ($D,E$) conditions reconstructed by SM2R and PMERGE-D are by far longer than those obtained from IMD (Fig. 8a, Table 1).

Given the vastness and variety of the Indian territory, the thresholds defined in this work (Figs. 8a,b) do not claim to be used in local landslide early warning systems. To calculate trigger thresholds based on local homogeneous sub-areas would require a by far larger amount of data (Peruccacci et al., 2012). Here, the use of thresholds at wider regional scale aims at comparing the performance of satellite- and ground-based rainfall data in order to identify those products, which are suited to give robust landslide forecasts. A further improve would be the same analysis at local scale, in order to highlight which product works

better or worse in a given sub-area. As stated above, this also requires to enrich the landslide catalogue.

The ROC analysis (Fig. 9) shows that in India the products that work best are the hourly-based PMERG-H and IMERG-ER. A similar performance is also achieved with PMERG-D, while among satellite products, SM2R is the least performing. Overall, the efficiency of the ground-based IMD is the lowest in predicting the rainfall-induce slope failures. This outcome could be somewhat unexpected since IMD is obtained by the interpolation of point rainfall measurements from rain gauges

and, hence, we would expect more precise rainfall data than the spatially averaged (pixel scale) estimates of the satellite rainfall products. As an example, Brunetti et al. (2018a) in Italy found a better performance of the ground-based rainfall data compared to satellite data. We hypothesize here that the 25 km grid spacing (Fig. 6) is able to explain a possible underestimation of the rainfall in many areas of India where the rain gauge density is low or even where the stations are not properly working. This would also explain the larger dispersion of the IMD rainfall ($D,E$) conditions with respect to SM2R and PMERG-D (Fig. 8a).

Conversely, the other products have significantly higher spatial resolutions, i.e. 10 km for IMERG-ER and 12.5 km for SM2R, PMERG-D and PMERG-H. In addition to this issue, Gariano et al. (2020) observed that the use of daily resolution data leads to a general worsening of the forecasting capability compared to hourly data.

Inspection of Figure 8b reveals that the rainfall ($D,E$) conditions that have triggered the majority of the landslides listed in the catalogue (58% and 64% for IMERG-ER and PMERGE-H, respectively) have a rainfall duration less than 24 h. As a





consequence, the use of daily resolution data alters the actual triggering rainfall of those landslides. For SM2R and PMERG-D datasets, only a 7% of rainfall ($D,E$) conditions have a duration $D < 7$ days (IMD median value), and only 1% equal to one day. This is ascribed to the nearly continuous rainfall detected by SM2R and PMERG-D, especially in the monsoon months (Fig. 7b). For this reason, it would eventually be more appropriate to implement landslide early warning systems that use rainfall thresholds defined with hourly data.

This work represents a pioneering attempt to ascertain at the regional (sub-continental) scale of India which are the best products for the forecast of rainfall-induced landslides. The results suggest that among the available satellite and ground-based products, the best performing ones are those meeting an hourly temporal resolution with an adequate spatial sampling.

## 6      Code availability

The CTRL-T software tool was written using the R open-source software and can be freely downloaded at:
https://doi.org/10.5281/zenodo.4533719.

## 7      Data availability

The SM2RAIN-ASCAT rainfall dataset is available for free at:
https://doi.org/10.5281/zenodo.3972958.
The GPM IMERG Early Run dataset is available after registration at:
https://search.earthdata.nasa.gov/search?q=GPM_3IMERGHHE_06.
The IMD dataset is provided by the Indian Meteorological Department.

## 8      Author contribution

MTB, MM, SLG and SP designed the experiment and carried it out. LC, LB and GA provided the rainfall data, MTB and SP collected the landslide information, and MM performed the simulations. MTB, MM, SLG, LB and SP analysed the results and
MTB, MM, SLG, and SP prepared the figures. All authors wrote and revised the text.

## 9      Competing interest

The authors declare that they have no conflict of interest.



## 10  Acknowledgements

We gratefully acknowledge support from EUMETSAT through the Global SM2RAIN project (contract no. EUM/CO/17/4600001981/BBo) and the "Satellite Application Facility on Support to Operational Hydrology and Water Management (H SAF)" CDOP 3 (grant no. EUM/C/85/16/DOC/15). We thank the Indian Meteorological Department (IMD) for providing the ground-based data and the Geological Survey of India for making landslide information accessible on their official website www.gsi.gov.in.

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



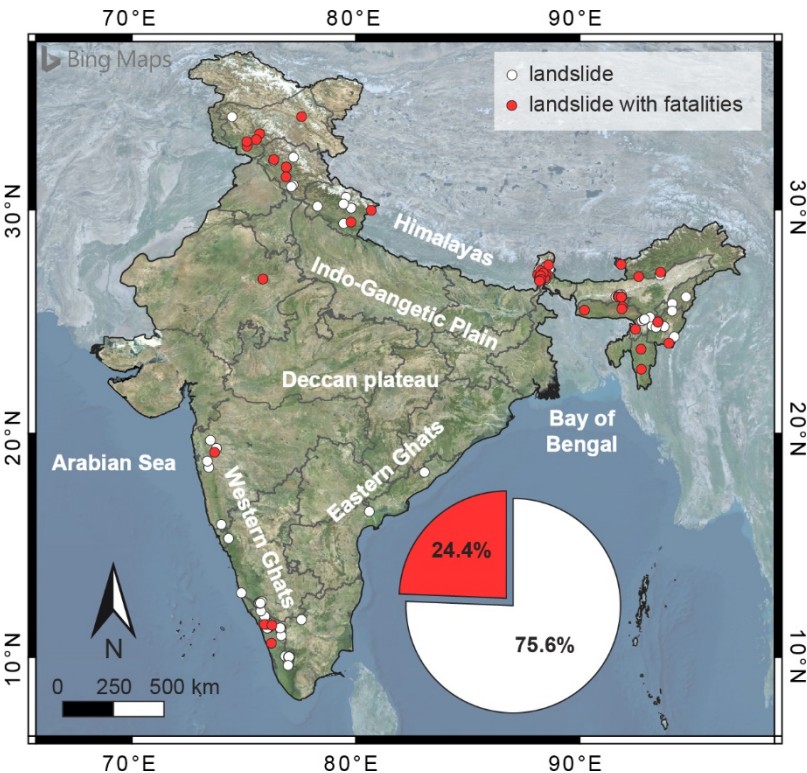

**Figure 1: Map of India with the location of landslides triggered by rainfall. Red and white dots highlight the slope failures with and without associated fatalities, respectively. Background image from Bing © Microsoft.**



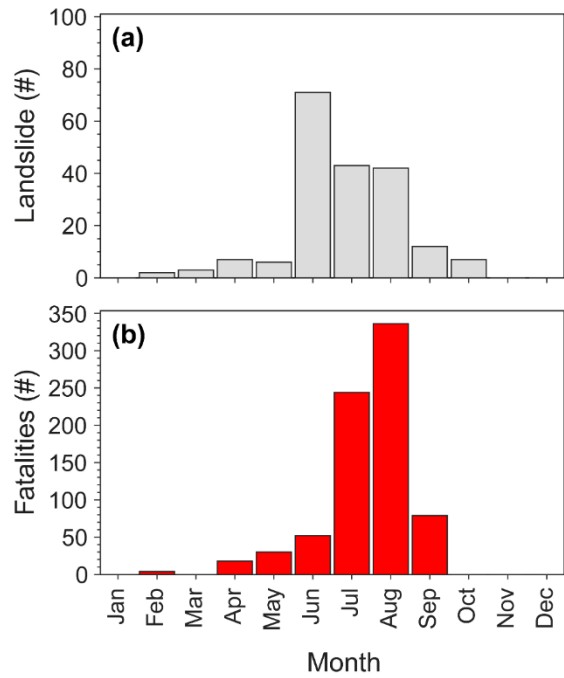

**Figure 2: Monthly distribution of (a) landslides and (b) landslide fatalities.**





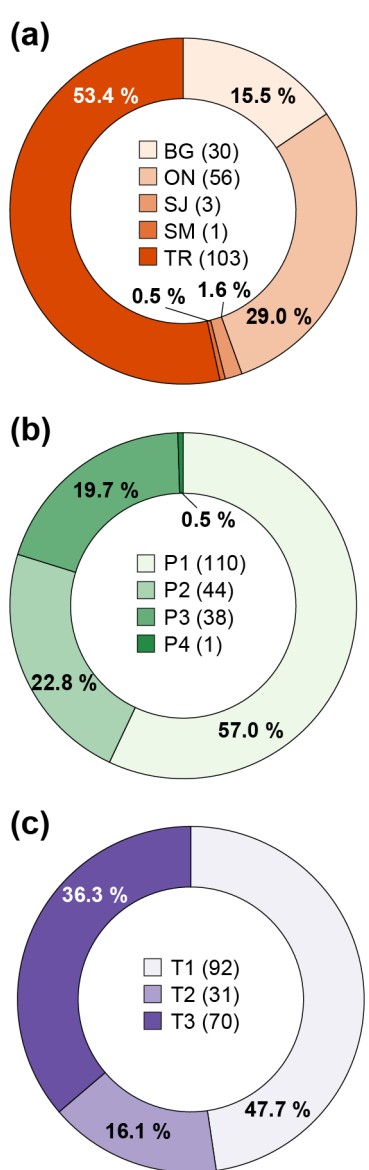

**Figure 3: Donut charts with statistics of the landslide catalogue. (a) Number and percentage of landslide information source; key: ON, online newspapers and magazines; BG, blogs; TR, technical reports; SJ, scientific journals; SM, social media; (b) number and percentage of landslides with decreasing mapping accuracy; key: P1 < 1 km² (high), 1 km² ≤ P2 < 10 km² (medium), 10 km² ≤ P3 < 100 km² (low), 100 km2 ≤ P4 < 300 km² (very low); (c) number and percentage of landslides with decreasing temporal accuracy; key: T1, hour of occurrence known; T2, part of the day known; T3, only day of occurrence known.**




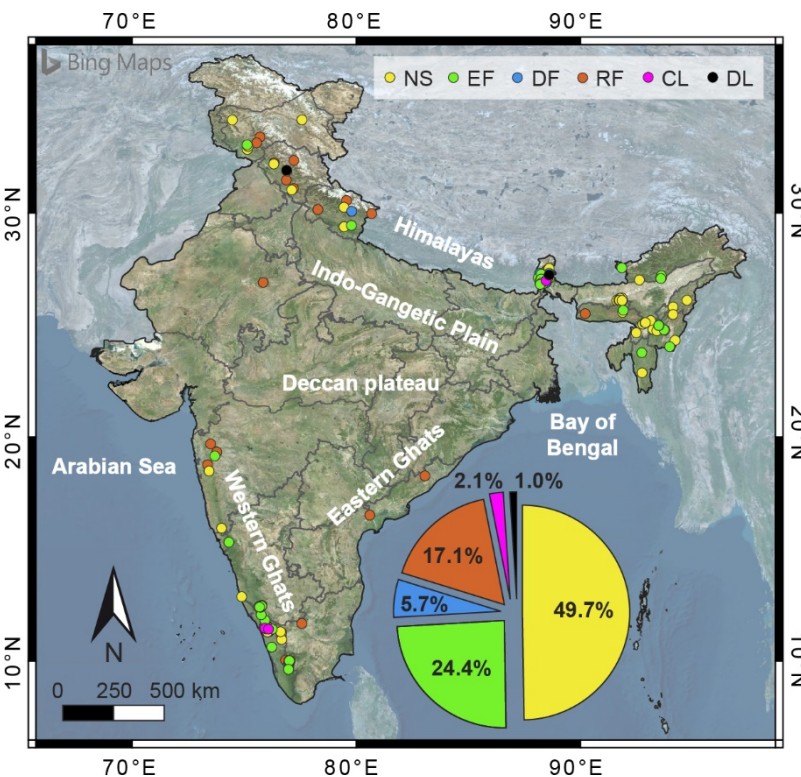

**Figure 4: Map of the location of the six types of landslide and the relative percentages. Key: CL, complex landslides; DF, debris-flow; EF, earth or mud flow/slide; RF, rock fall/slide; DL, unspecified deep-seated landslide; NS, not specified landslide. Background image from Bing © Microsoft.**



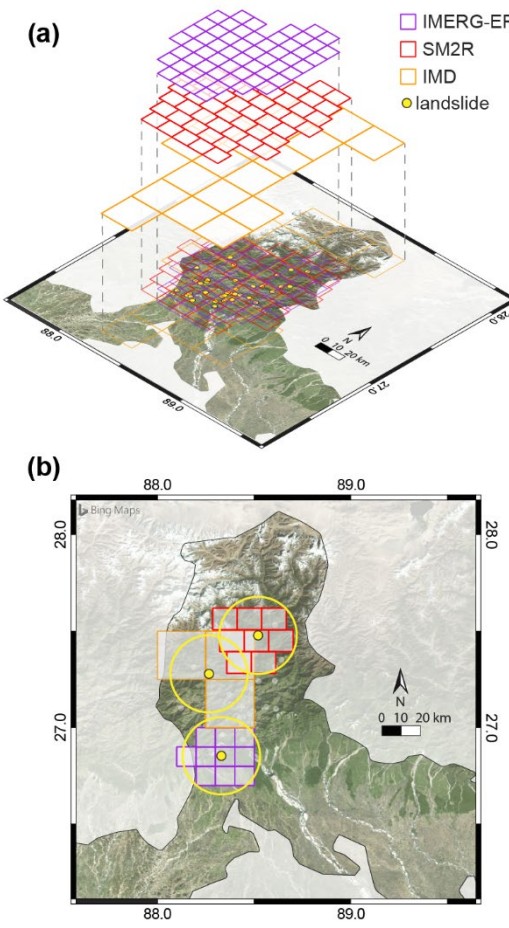

**Figure 5: (a) Comparison of the pixel resolution for the satellite-based IMERG, SM2RAIN and the ground-based IMD rainfall products. (b) Example of the pixel selection for three landslides using a 20 km buffer (yellow circle) for the state of Assam. Background image from Bing © Microsoft.**


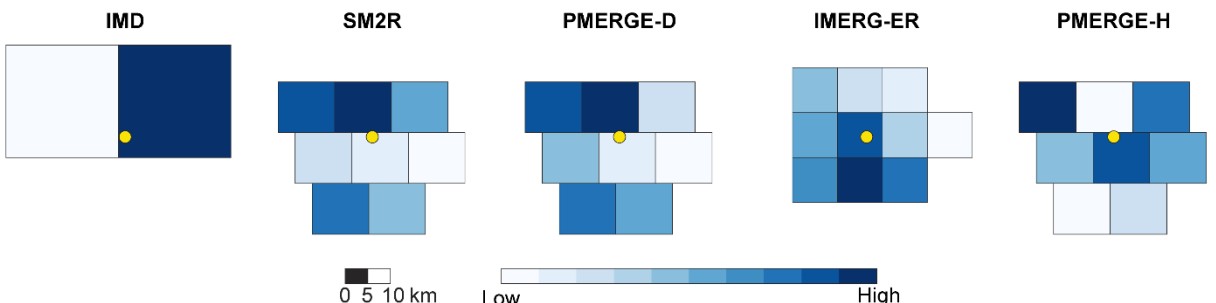

**Figure 6: Pixels enclosed in the buffer of a landslide (yellow dot) for each rainfall product coloured based on the weight. The representative pixel is the darkest one. The colour scale is qualitative, since it is different for each product.**

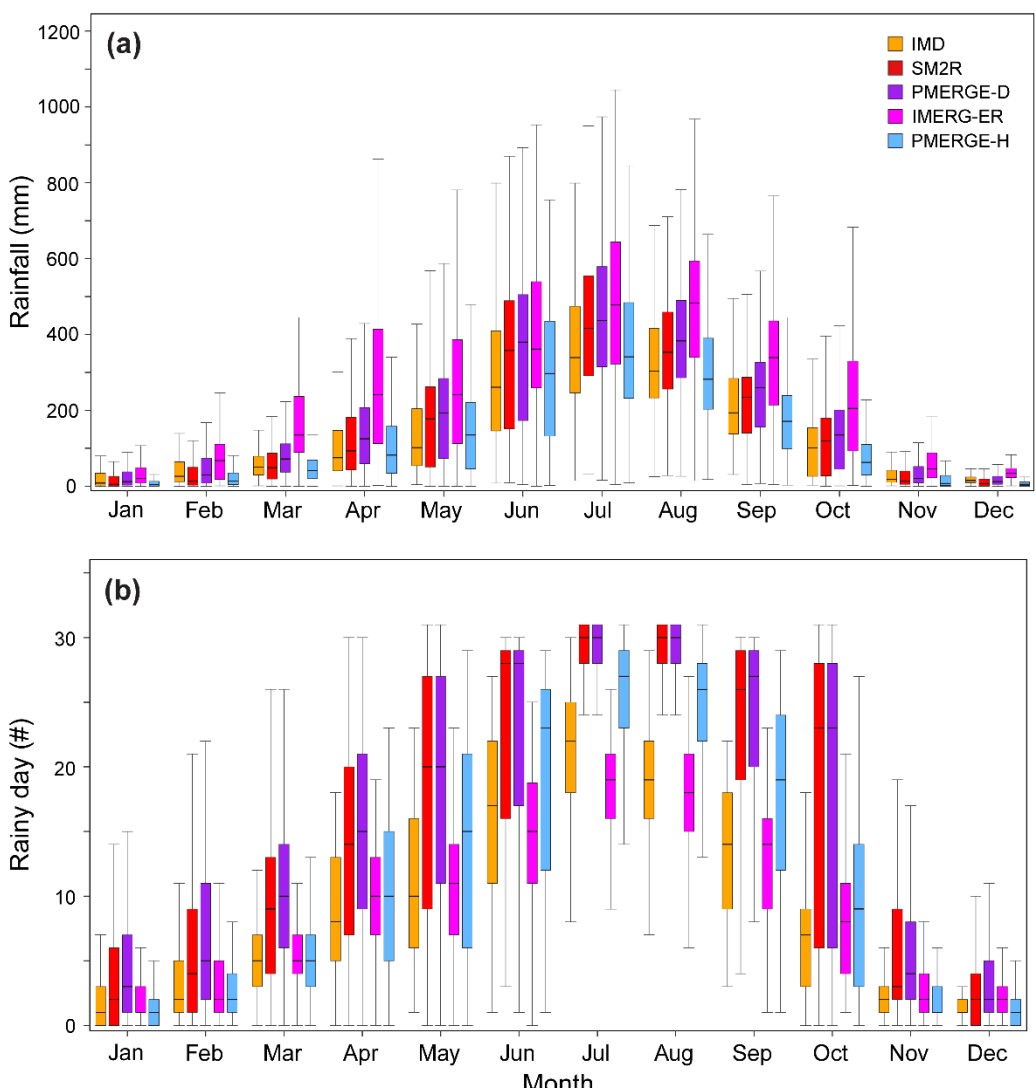


**Figure 7: Box-and-whiskers plots showing the distribution of (a) monthly amount of rainfall as estimated by the satellite-based and measured ground-based products in the selected pixels, and (b) monthly number of rainy days.**





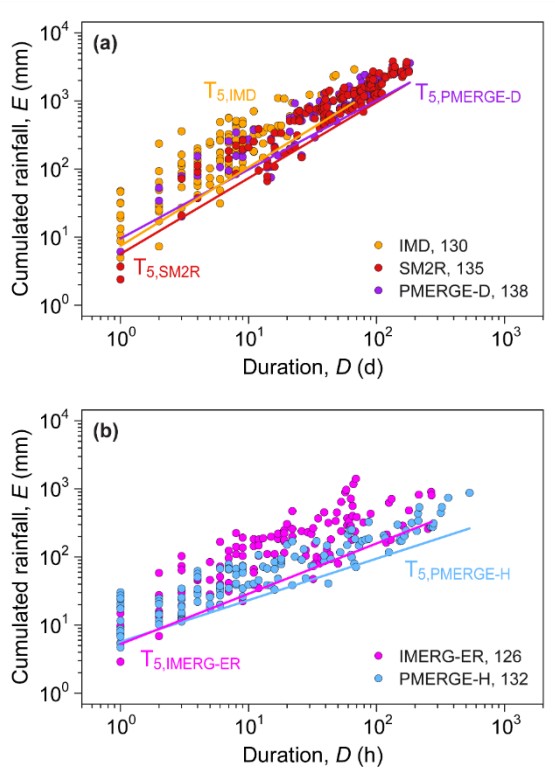

**Figure 8: Rainfall ($D,E$) conditions and rainfall thresholds at 5% NEP level for (a) the daily resolution products IMD, SM2RASC and PMERGE-D, and (b) the hourly resolution products IMERG and PMERGE-D.**



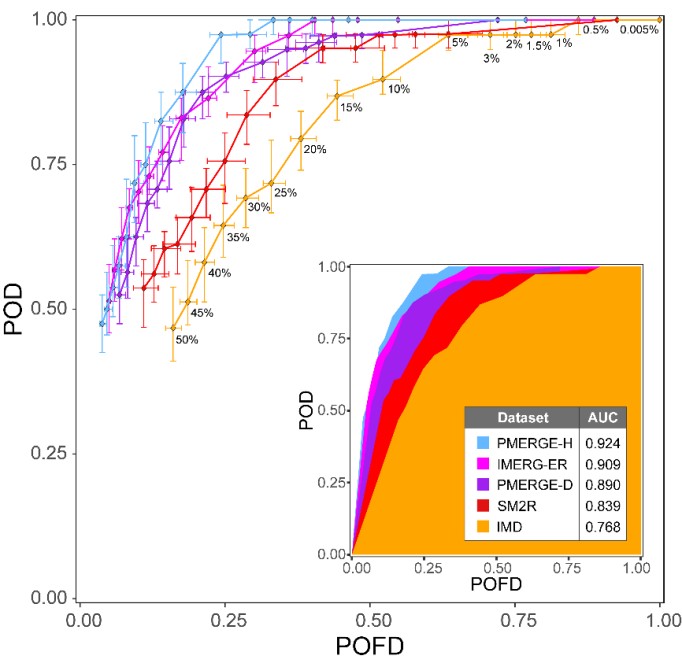

**Figure 9: ROC analysis derived varying the NEP for IMD (orange), SM2R (red), PMERGE-D (purple), IMERG (magenta) and PMERGE-H (blue) data sets. Error bars depict the interval of variation of POFD and POD. Inset graph shows AUC for each product.**



**Table 1. ED thresholds for possible landslide occurrence in India. Label identifies the thresholds established in this work for the listed product. NL, number of landslides with reconstructed rainfall conditions. *D* and *E*, the rainfall duration (in days or hours) and cumulated event rainfall (in mm). Threshold column lists the equations for the 5% *ED* thresholds.**


| Label | Product | $N_L$ | $D$ (d) | | | $E$ (mm) | | | Threshold |
|---|---|---|---|---|---|---|---|---|---|
| | | # | min | median | max | min | median | max | |
| $T_{5,IMD}$ | IMD | 130 | 1 | 7.0 | 67 | 5.0 | 181.8 | 2912,0 | $E = (7.4 \pm 1.2) \cdot D^{(1.1 \pm 0.06)}$ |
| $T_{5,SM2R}$ | SM2R | 135 | 1 | 52.0 | 179 | 2.4 | 823.5 | 3813.8 | $E = (5.7 \pm 1.6) \cdot D^{(1.1 \pm 0.06)}$ |
| $T_{5,PMERGE-D}$ | PMERGE-D | 138 | 1 | 52.5 | 182 | 6.1 | 938.6 | 3609.3 | $E = (9.5 \pm 2.0) \cdot D^{(1.0 \pm 0.04)}$ |

| Label | Product | $N_L$ | $D$ (h) | | | $E$ (mm) | | | Threshold |
|---|---|---|---|---|---|---|---|---|---|
| | | # | min | median | max | min | median | max | |
| $T_{5,IMERG-ER}$ | IMERG-ER | 126 | 1 | 17.0 | 269 | 2.9 | 143.1 | 1413.4 | $E = (5.3 \pm 0.8) \cdot D^{(0.74 \pm 0.04)}$ |
| $T_{5,PMERGE-H}$ | PMERGE-H | 132 | 1 | 11.5 | 533 | 4.7 | 66.8 | 874.7 | $E = (5.7 \pm 0.5) \cdot D^{(0.61 \pm 0.02)}$ |