# Peer review of "Satellite rainfall products outperform ground observations for landslide prediction in India"

_Hydrology and Earth System Sciences, 2021_

## Referee Comment (RC1)

[referee-annotated manuscript omitted]

---

## Author Comment (AC2)

Replies to comments from reviewer #1

We thank the Reviewer #2 for his/her kind comments.

**Line 9: Remote sensing is not a tool. Please rephrase the sentence**

R: We rephrased with: "*In these areas, remote sensing can represent an important detection and monitoring process to predict landslide occurrence...* ".

**Line 10: Is it forecasting? It seems like prediction. There is difference between prediction, forecast and hindcasting. Please be clear what is actually being done**

R: You're perfectly right. Indeed, here we perform a prediction of rainfall-induced landslides.

**Line 15: Any specific reason to use only these products? Why not check other climatic reanalysis datasets?**

R: We decided to test these two products as we wanted to highlight the added value of using satellite data. The choice of GPM and SM2RAIN-derived rainfall lies in the fact that we wanted to test the capabilities of two different retrieval approaches, i.e. a state-of-the-art classical satellite rainfall product and a novel technique that estimates rainfall through soil moisture observations, to provide useful information for landslide applications. Moreover, we decided to not consider other climatic reanalysis data as we wanted to highlight the feasibility of using rainfall estimates in available in near-real time for operational purposes. Indeed, GPM data are provided within 4 hours after sensing (the Early run product here considered), a near-real time product based on SM2RAIN is currently under development within HSAF, while ERA 5 for instance is provided 5 days after the model run.
We better clarify this point at lines 188-191:
"*The satellite rainfall products here considered allow to test the capabilities of two different retrieval algorithms, i.e. a state-of-the-art classical satellite rainfall product and a novel technique that estimates rainfall through soil moisture observations. Other reanalysis data were not considered as we wanted to investigate the use of near-real time satellite products for a possible future operational application.*"

**Line 19: Most of the landslide data in this region is on a daily basis. How did you find the hourly based landslide location. Was the historical landslide events collected based on high temporal resolution satellite images?**

R: We didn't use satellite images to locate the landslides. We collected landslide information from online newspapers and magazines (ON), blogs (BG), technical reports (TR) made available by the Geological Survey of India (GSI, www.gsi.gov.in), scientific journals (SJ) and social media (SM). For the majority (about 64%) of the landslides we found the occurrence time or the part of the day.

**Line 34: Too many references, for one generic statement. Either mention the country and the corresponding citation or mention only the notable review papers.**

R: We removed the following references: Ruiz-Villanueva et al. 2011, Rosi et al., 2016, Peruccacci et al. (2017), but we prefer to leave the others.

**Line 59: You can also mention the recent work of Dikshit et al. (2020) where they argued the necessity to use satellite based products in the Indian Himalayan region.**
**Dikshit, A. et al. Rainfall Induced Landslide Studies in Indian Himalayan Region: A Critical Review. Appl. Sci. 2020, 10, 2466.**

R: We added: "*In a recent work, Dikshit et al. (2020b) argued the necessity to use satellite-based products in the Indian Himalayan region.*"

**Line 76: See, here the word "predict" is used. Please be thorough in the entire manuscript.**

R: Done.

**Line 76: English problem.**

R: We have modified the sentence as follows: "*we collected information on rainfall-induced landslides in a new catalogue.*"

**Line 99: You can also add the Koppen climate classification.**

R: It could be interesting to use the Koppen classification to investigate the role of the climate. In India there are at least 5 Koppen climate classes, but the limited number of landslides available for this work prevents such analysis.

Line 101: What do you mean by statistically significant? Do you mean total number of landslides in all of India or total number in a specific region? If so, how many and at what area?

R: Here, "statistically significant" means larger than a minimum number of landslides to define reliable regional thresholds as indicated in Peruccacci et al. (2012). In that work we found that this number is between 75 and 100.

Line 111: Can you add a list of landslide events included and discarded in the analysis?

R:  Definitely, we can share the catalogue, but the discarded events were not included in it, since they were not useful for our purposes. Events that did not have the required spatial and temporal accuracy were not stored.

Line 118: I don't think Rajasthan has a problem of landslide occurrences. Kindly check.

R: Unfortunately, this unique landslide killed one and injured six residents in the foothill zone of Amagarh Hill.

Line 121: Could you elaborate on this?

R: We changed "*social incidence*" in "*impact on the population*".

Line 122: I assume that this finding was based after 50% of the landslide data was discarded. If they were not discarded, does it depict the same results?

R: As we stated in a previous answer, the discarded events were not stored and therefore were not available for this analysis.

Line 125: Please check Uttarakhand landslides should be a big contributor on fatalities

R: What is shown in Fig. 2b is relative to our catalogue, which was compiled with the criteria described above. In Uttarakhand, we had many problems in finding the landslide coordinates, since many of the failures occurred "along" long routes, and were consequently discarded from the catalogue.

Line 128: Were these reliable, in terms of location and dates. Did you cross-check it with newspapers?

R: As per fig. 3a, social media accounted only for 0.5%, and they were checked with newspapers. Social media are especially useful to find details on the time of occurrence of landslides.

Line 135: Does this sort of classification have an effect on the model outcomes or is it providing an analysis on the landslide type in the region

R: Due to the limited number of landslides in each class, the classification is only providing an analysis on the landslide type.

Line 150: Reason for using this interpolation? Why not krigging?

R: We used the IMD product as it is provided by the Indian Meteorological Department, we did not perform any other processing step to ground data.

Line 156: for which region was these studies conducted?

R: Europe.

Line 183: Here, the integration and the way rainfall is estimated is not clear. Although, you provide references that these data and processes seem to work, was it effective for determining rainfall in Indian context and if they were what were the uncertainties in different parts of the country.
This brings to my previous comment, why not use climatic reanalysis dataset like ERA5, by that way you could have processed more data points?

R: In order to highlight the fruitful effect of the integration between the two datasets, we performed a very simple test, estimating the daily correlation coefficient and RMSE, using ERA5 as benchmark. As a result, the performance of the integrated product improved both in terms of correlation and RMSE on a daily scale, throughout the study area. The results can be observed in the following figures.

[Figure]

For what concerns ERA 5, please refer to our previous comment.

Line 201: The buffer of 20km was it used for entire country or was it used in specific parts of the country? Why not put a varying buffer size dependent on the rain gauge density.

R: We applied the same buffer radius all over the country. For comparison with gridded satellite data, we used gridded ground rainfall data provided by IMD, which doesn't contain any information on the rain gauge density.

Line 250: English problem.

R: We rephrased it as follows: "*These two products exhibit the highest number of rainy days along the whole year*"

Line 277: After reading this section, it looks that only the rainfall parameter is being predicted, but not the locations? Thus, the title and the wording in the manuscript should change to "prediction of rainfall thresholds for landslide occurrences" rather than "forecasting of landslides"

R: We agree, and we changed the title and the wording in the manuscript.

Line 325: Use a milder word. I understand this is an excellent work, but I don't think pioneering is the right word.

R: Done.

Figure 2: Please mention the years.

R: Done.

Figure 7: If you could use different colors for better clarity, that would be appreciated.

R: We prefer to keep these colors.

---

## Author Comment (AC3)

Replies to comments from Ben Mirus

This paper presents the first examination of different rainfall products for landslide forecasting in India. The products tested includes a variety of satellite products with a range of spatial resolutions and daily or hourly temporal resolutions, which are compared to a ground-based rainfall product at relatively coarse spatial resolution and daily temporal resolution. The testing relies on an extensive, high-quality landslide inventory database across different regions of India, which provides a suitable foundation for the conclusion that the higher resolution, hourly satellite products provide the best performing rainfall ED thresholds. Not surprisingly, the ground-based database performs poorest when evaluated with standard ROC metrics since the spatial and temporal resolution is inferior.

The paper provides a useful approach and template for future comparisons across different regions, and additionally the conclusions are also useful for application and testing of more detailed, regional rainfall thresholds across India. Overall the paper will be of interest to readers of HESS. There are some important questions and details related to the ground-based dataset, which need to be provided and discussed to provide a complete analysis of the results and conclusions. I also recommend a few references and discussion points that would provide fuller context of this contribution, since not all datasets are equal or equally applied and considered across the world. Otherwise, following these revisions listed below by line number, the paper should ultimately be published in HESS.

R: We sincerely thank Ben Mirus for his useful comments and suggestions.

130. This is very intriguing, but also a bit unclear. Could you provide a little more detail about the accuracy classification?

R: We agree and we modified the sentence as follows: "*Each landslide in the catalogue is assigned a position accuracy level in four classes (Figure 3b): P1 < 1 km$^2$ (high), 1 km$^2$ ≤ P2 < 10 km$^2$ (medium), 10 km$^2$ ≤ P3 < 100 km$^2$ (low), and 100 km$^2$ ≤ P4 < 300 km$^2$ (very low). Similarly, a decreasing temporal accuracy in three classes is associated to each landslide (Figure 3c). The first class (T1) collects the events for which the time of occurrence is known with an accuracy of one hour, while for the second (T2) and the third (T3) classes the part of the day is inferred or the day of occurrence is known (Peruccacci et al., 2012).*". As an example, a level P1 was attributed to a landslide mapped with a geographic accuracy of 1 km$^2$ or less (a radius less than about 0.6 km) as reported in the cited reference.

We introduced a semi-quantitative "confidence" metric in the U.S. inventory (Mirus et al., Landslides, 2020, https://doi.org/10.1007/s10346-020-01424-4), which is used to reflect overall confidence in information source and detail (position, nature and extent, actual occurrence of landslide). However, our approach was rather ad-hoc based on subjective expert opinion. I like your idea of splitting the temporal and spatial accuracy and so I (and probably other readers) would also like to learn more.

R: Thanks for the question, which opens up an interesting issue, which is the reliability of rainfall-triggered landslide data. In our experience, the temporal and spatial accuracy of landslides are almost never correlated. Our catalogue is mainly done of landslides that impacted people or infrastructure. Sometimes the location is very detailed (e.g., "highway at km. 121, near the temple of …", or "in the village of Chamoli") and the time is more uncertain (e.g., "yesterday", or "in the morning"). Other times, especially in case of deaths, the occurrence time may be very detailed (e.g., "at 2:30 AM"). We found that landslides in GSI reports are usually well located (P1), since they contain longitude and latitude.

Splitting temporal and spatial accuracy is fundamental to determine rainfall conditions able to trigger landslides in different environmental setting (e.g., lithological, topographic, etc.) for which a "finer" spatial resolution is needed.

Specifically, in looking at Figure 3 it seems your rating has more to do with precision/resolution, rather than accuracy, at least for the timing (3c), but if the exact location of the landslide is unknown it is unclear how did you determine the spatial accuracy (resolution?) component.

R: Figures 3b and c portray the mapping accuracy with which landslides were spatially and temporally located, respectively. The term "resolution" in this work refers only to rainfall data, i.e. the spatial resolution is the pixel size and the temporal resolution is hourly or daily.

Please add one or two sentences for each accuracy classification to describe how it was developed and assigned, and be sure to clarify whether you mean actually "accuracy" in both cases or if some other description is more accurate.

R: We should have answered to this question above.

149-153. While I recognize that this is an impressive number of rain gages and there must be some variability between them, it would be useful to describe what general type of rain gages (i.e., tipping bucket ormanual?) and the typical location of the gages (i.e. in cities/towns/airports?). If there is not pattern, perhaps some general description of the objectives of the Indian government in deploying this network.

R: India maintains a large network of Automatic Weather stations (AWS) and Automatic Rain gauge stations (ARG) for near real-time weather forecasting including the manual measurements across different locations from cities to airports. However, there are still areas with high-terrain and complex landslide locations, where installation and maintenance of rain gauges is hard. Right in these areas, satellite-based rainfall measurements might provide great opportunity for landslide early warning. We don't have more specific information on the general type of these sensors. As reported in lines 149-151, IMD provided us a gridded dataset with a spatial resolution of 0.25° over the territory of India.

While the satellite measurements will have bias based on the measurement techniques, the pixel averaging of the ground-based measurements will be greatly influenced by the type and location of these point measurements, so some understanding of this would be helpful.

R: This is a very interesting topic indeed, since it can be applied everywhere and not only in India, but is beyond the scope of this paper.

159&162&183. Why is 0.1 degree listed as 12.5km, 10km, and then 12.5km again? Presumably there is some variability over the study area, but if so the appropriate range or average should be specified.

R: You are right, there is obviously some variability depending on the latitude, so now we give throughout the text the spatial resolution in degrees.

114-199. This process is not entirely clear. Since this is not a standardized calibration approach, a little more detail is necessary to understand and repeat this process. Also, how was the 5mm median determined?

R: We now explained it in more detail, and we modified the sentence as follows: "*The comparison between hyetographs from ground-based and satellite-based products highlighted that the remote sensing signal was found even though it was not actually raining, and it was independent of the satellite product and of its temporal resolution. Therefore, we set heuristically a filter that removes 2 mm per hour for IMERG-ER and PMERG-H and 2 mm per day for IMD, SM2R and PMERG-D. We verified that the filter removes a small percentage (5 mm as a median) of the daily cumulated rainfall, nonetheless this amount is nearly irrelevant for the landslide triggering.*"

222. Can you provide slightly more detail about the 100 synthetic series? For example, are these synthetic time series of a potential landslide-inducing rainfall event?

R: We thank you for this question, because we found a mistake in the text. We rephrased this part as follows: "*In order to assess the performance of the ground- and satellite-based rainfall products*

*in predicting landslides in India a validation of the thresholds has been carried out, following Brunetti et al., (2018a). For each product, we sample randomly without repetition 70% of the reconstructed (D,E) pairs to obtain 100 new datasets. Then, for each dataset we calculate rainfall thresholds at varying NEP. The remaining 30% of (D,E) pairs is used to evaluate the threshold performance by means of contingency table (reporting binary classifiers of rainfall conditions that triggered or did not trigger landslides), skill scores and ROC (Receiver Operating characteristic) analysis, following Gariano et al. (2015) and Brunetti et al. (2018a).*"

276-277. Yes, but some further discussion of this acknowledgement is needed in the following section.

R: We agree with you. The relative discussion was indeed in the following section, lines 306-317. Thanks to your suggestions, we have now integrated some additional considerations (see below).

304. Typo: improvement

R: Done.

306. Yes, we also found this (Thomas et al., WRR, 2019, https://doi.org/10.1029/2019WR025577) in which a network of 96 tipping buckets (~2.5km spacing) was more accurate at capturing the orographic influence on rainfall intensity and cumulative amounts, when compared to the IMERG dataset. In our case the density of gages is much higher and includes measurements across a range of elevations relevant to landslide initiation. I suspect that the IMD used in your study does not capture rainfall measurements in steep, landslide-prone terrain, which may further contribute to the underestimate. Thus the degree to which ground-based measurements out- or under-perform relative to satellite products may depend very heavily on where the in-situ measurements are placed within the landscape.

R: We agree with you, and we modified the text as follows "*Thomas et al. (2019) in the San Francisco Bay Area, using a network of 96 tipping buckets (~2.5 km spacing), found that the ground-based dataset was able to capture the orographic influence on rainfall intensity and cumulated amounts more accurately than the IMERG dataset. In our case, we hypothesize that the 0.25° grid spacing (Fig. 6) is able to explain a possible underestimation of the rainfall in many areas of India where the rain gauge density is low or even the stations are not properly working. Moreover, the degree to which ground-based measurements out- or under-perform relative to satellite products may depend on where the in-situ measurements are placed within the landscape especially in steep, landslide-prone terrain. The above considerations would also explain the larger dispersion of the IMD rainfall (D,E) conditions with respect to SM2R and PMERG-D (Fig. 8a).*"

315. For clarity, consider stating "finer" spatial resolution. I have found that some people mis-interpret "higher" resolution as "coarser" resolution.

R: Done.

317. I think it's worth noting here that Leonarduzzi and Molnar (NHESS, 2020, https://doi.org/10.5194/nhess-20-2905-2020) found that the daily vs. hourly resolution is not always so clear, but depends also on the temporal resolution of the landslide inventory. In your case your inventory has fairly precise timing information (Figure 3c). This relates directly to your next point about the duration of landslide triggering events as well.

R: We maintain that the result obtained by Marra (2019) and Gariano et al. (2020) is always valid. The database of hourly rainfall measurements used by Leonarduzzi and Molnar (2020) covers a limited time period (recent years only) and their hourly rain gauge network has a low density.